# Ion Transport and Electrochemical Reaction in LiNi_0.5_Co_0.2_Mn_0.3_O_2_-Based High Energy/Power Lithium-Ion Batteries

**DOI:** 10.3390/nano13050856

**Published:** 2023-02-25

**Authors:** Jinmei Xu, Jiandong Yang, Shaofei Wang, Jiangmin Jiang, Quanchao Zhuang, Xiangyun Qiu, Kai Wu, Honghe Zheng

**Affiliations:** 1College of Energy, Soochow University, Suzhou 215006, China; 2Contemporary Amperex Technology Co., Ltd., Ningde 352100, China; 3School of Material and Physics, China University of Mining and Technology, Xuzhou 221116, China; 4Power & Energy Storage System Research Center, College of Mechanical and Electrical Engineering, Qingdao University, Qingdao 266071, China

**Keywords:** NCM523 HEP LIB, § ≡ *R*_ct_/*R*_ion_, de Levie-Finite Length Pore-*L*_s_

## Abstract

The high energy/power lithium-ion battery using LiNi_0.5_Co_0.2_Mn_0.3_O_2_ (NCM523 HEP LIB) has an excellent trade-off between specific capacity, cost, and stable thermal characteristics. However, it still brings a massive challenge for power improvement under low temperatures. Deeply understanding the electrode interface reaction mechanism is crucial to solving this problem. This work studies the impedance spectrum characteristics of commercial symmetric batteries under different states of charge (SOCs) and temperatures. The changing tendencies of the Li^+^ diffusion resistance *R*_ion_ and charge transfer resistance *R*_ct_ with temperature and SOC are explored. Moreover, one quantitative parameter, § ≡ *R*_ct_/*R*_ion_, is introduced to identify the boundary conditions of the rate control step inside the porous electrode. This work points out the direction to design and improve performance for commercial HEP LIB with common temperature and charging range of users.

## 1. Introduction

With the increasing demand for driving range improvements of new energy vehicles, ternary cathode materials have received wide attention. Compared with NCM111, LiNi_0.5_Co_0.2_Mn_0.3_O_2_ (NCM523) has a higher specific capacity, lower cost, and similar thermal stability, making it the largest market share in the current ternary material products. Although the content of Ni in NCM523 is relatively high (up to 50%), most of these materials exist in the form of Ni^2+^, and the proportion of Ni^3+^ is only 20%. Therefore, NCM523 is a typical medium–nickel-type material that can be prepared in the standard air atmosphere, similar to LiCoO_2_ and NCM111.

After nearly ten years of process optimization, various types of NCM523 products for different market segments have been successively developed, and the maturity and stability of productive processes have been continuously improved. As a result, lithium-ion batteries using NCM523 are widely used in notebook computers, power banks, power tools, electric bicycles, hybrid electric vehicles, pure electric vehicles, and many other products. After years of verification in the market, although pure electric vehicles do not have as high a power-density requirement as hybrid electric vehicles for lithium batteries, they still require a power density of close to 2000 W·kg^−1^ in practical application. 

On the other hand, the state-mandated vehicle warranty for more than one year also considers the public and private use characteristics of a considerable proportion of pure electric vehicle owners. Therefore, pure electric vehicles with lithium-ion batteries should have an ultra-long cycle life. Generally, the warranty for vehicles in public service has increased to a life cycle of 600,000 to 1,000,000 km, requiring high power and energy densities for lithium-ion batteries used for pure electric vehicles, along with the ability to withstand high-frequency use. In view of the long-term cycle, NCM523, with single crystals and small particles suitable for such scenarios, is currently the most widely used material in the market.

Unfortunately, most of the research on Li_x_Ni_1−y−z_Co_y_Mn_z_O_2_ in academia focuses on using different material synthesis methods and other functional materials to modify the ternary cathode to improve its gram capacity and electrical conductivity and optimize the electrolyte to match the electrochemical aspects, such as long-term cycle performance. Other research concerns include the surface coating of TiO_2_ to improve the intrinsic safety of materials [1,2,3,4] and how to efficiently repair such NCM523 high energy/power lithium-ion batteries (NCM523) with high recovery values [5,6]. Another area is simply changing the thickness or porosity of the electrode sheet to study the changing trend of its magnification characteristics. However, relatively few studies have focused on the porous electrode characteristics of such commercial single-crystal small-particle NCM523 materials. The characterization techniques used in the aforementioned research are relatively simple, such as cyclic voltammetry (CV) and the conventional electrochemical AC impedance method [7]. 

According to the US DOE 2003 report [8], there is a quantitative relationship between the power characteristics of the battery and the internal resistance of the battery, Therefore, the main work focuses on determining the key factors affecting the internal resistance of the battery and effectively regulating them. LIB electrode films are generally typical porous electrode systems. Exploring of the basic theory of porous electrodes, analyzing and decomposing the source of battery internal resistance is the first step in the research work. EIS is a non-destructive testing tool widely used in resistance analysis and the analysis of test spectrum is also mature. Additionally, EIS technology can explain which resistance dominates the process of lithium-ion intercalation and determine the key to the resistance. Moreover, this method has become a popular tool for researchers to study interface reactions. Researchers have carried out several studies on the sources of internal resistances, which rely on various parameters, such as temperature or over-potential [9,10,11,12,13,14]. However, the electrode preparation in the related research work was mostly completed in the laboratory. In addition, the obtained electrode often contained a large fraction of auxiliary materials, such as adhesive and conductive carbon (>5 wt% for each). Furthermore, homogenizers frequently used in laboratories would obtain an electrode film with a thickness of 100 to 200 µm. As a result, uneven mixing would occur, and uncontrollable factors would be introduced, making the process quite different from the industrial LIB electrodes. Therefore, the results of these academic studies can only provide qualitative and ideal mechanism understanding, and the quantitative guidance for industrial product design optimization is very limited.

Considering the above-mentioned points, this work proposes using NCM positive and graphite negative electrode sheets from NCM523 HEP LIB to assemble symmetric cells. The impedance spectrum characteristics of the symmetric batteries are studied under common temperature and state of charge intervals for electric vehicles. In order not to repeat the description, the following temperature and state of charge are hereby defined. SOC1, 2, 3, 4, 5, 6, 7 = 0%, 10%, 30%, 50%, 70%, 90%, and 100% separately, and Temperature (T1, 2, 3, 4) = −20, 0, 10, and 25 °C separately. A symmetrical battery is composed of pole pieces in exactly the same state, avoiding interference with the electrodes, which prevents the data from being parsed. The impedance data is fitted using the equivalent circuit established based on the cognition of electrochemical reaction mechanism [15,16,17,18]. The effects of temperature and SOC on lithium-ion liquid-phase diffusion resistance *R*_ion_ and charge transfer resistance *R*_ct_ in the porous electrode are discussed. Additionally, the related kinetic parameters are investigated in this work.

## 2. Materials and Methods

All electrode pieces used are obtained from the batch production delivery line, according to the design requirements of commercial power batteries. The battery size was standard VDA (Verband der Automobilindustrie) dimensions of 26.5 × 148 × 91 mm. The area of the cathode electrode was 2.6 m^2^. In addition, the NCM523 electrode was composed of 96 wt.% NCM523 (Tianjin Bamo Technology Co., Ltd., Tianjin, China), 1.5 wt.% adhesives (Arkema, France), and the remaining proportion is conductive agents (TIMCAL, Shanghai, China). The negative electrodes were prepared using 96 wt.% artificial graphite, 1 wt.% conductive carbon black, 3 wt.% sodium carboxymethyl cellulose, and water-based adhesive. The loadings of NCM523 and graphite electrodes were 8.3 and 4.8 mg·cm^−2^, respectively. The thicknesses of the positive electrode and negative electrode after cold pressing were 25 and 31 μm, respectively. The electrolyte was 1.0 mol·L^–1^ LiPF_6_ dissolved in a solution of EC:EMC:DEC (*v*:*v*:*v* = 3:5:2). The separator was polyethylene with a thickness of 7 µm (Shenzhen Xingyuan Electronic Technology Co., Ltd., Shenzhen, China). In the full battery, the current densities of the SOCs were regulated by the SOC at 0.1 C (where 1C is equal to 37 A, which is calculated from the following equation:total effective area of cathode electrode × coating weight × active material ratio × specific capacity test in coin cell = 2.6 m^2^ × 8.3 mg·cm^−2^ × 96 wt.% × 178.4 mAh·g^−1^).

Morphology images were investigated by the scanning electron microscope (Zeiss, Oberkochen, Germany). The materials were characterized by Cu Kα radiation using a Bruker D8 Discover X-ray diffractometer (XRD; Bruker, Germany). The scanning range of the NCM523 powder was 15° to 70°; the scanning range of graphite was 20° to 75°. The cell performances were tested using a 2430 button battery of Wuhan Blue Power 5V1mA 8CT (Wuhan Blue Power Electronics Co., Ltd., Wuhan, China). The voltage range of the NCM523 electrode test was 2.8 to 4.35 V during the charging and discharging processes. The current density was 0.1 C at 25 °C (where 1C = 2.2 mA). The voltage range of graphite electrode tests was 0.005 to 2.0 V, and the current density was set to 0.1 C at room temperature (where 1 C = 2.6 mA). 

The full battery charge–discharge performances were tested using the NMC523 HEP LIB, which were assembled in a hard case with a capacity of 37 Ah. The battery size was 26.5 × 148 × 91 mm, the test voltage range was 2.8 to 4.35 V, and the charging current was 37 A. The charge and discharge temperatures could be controlled using the high–low temperature box (SC 80 CC 3), and the test batteries were statically held for 6 h before the test. 

The EIS tests is conducted on a special spectrum acquisition instrument (VMP3, French), and scanning from 10^−2^ to 10^5^ Hz. The electrolytic cell adopted a symmetrical battery system, as shown in Figure 1. The symmetrical battery, a flexible packaging battery with an aluminum-plastic film, was 28 × 173 × 91 mm. Both the research electrode and energy-type lithium-ion battery NCM523 or graphite electrodes were under the fresh status, which means they would just perform one charge and discharge cycle and then be sent for disassembly. The symmetrical cell was externally fixed to a constant pressure control device to achieve good interface contact. The SOC of the symmetric battery could be changed by adjusting the SOC of the full battery, then the NCM523 and graphite electrodes were disassembled corresponding to the SOC, and finally used for symmetric battery assemblies. The full battery was disassembled after standing for 6 h to obtain electrode pieces for the symmetric batteries. 

## 3. Results and Discussion

The crystal structure test diagram of NCM523 is shown in Figure 2a. All reflections are in good agreement with the hexagonal structure that belongs to the space group R$\bar{3}$m, which is the same as layer-structured α-NaFeO_2._ The clear and strong diffraction peaks of 006/102 and 108/110 indicate that the layered structure of the material sample is good, and the positions of the characteristic peaks are consistent with the characteristic positions of the LiNiO_2_ in the XRD pattern. The value of I003/I104 is about 1.5, which is greater than the standard critical value of 1.2 (with the increase in the I003/I104 ratio, the chaotic occupation ratio of nickel ions entering the lithium slab decrease). This result indicates that the mixing of Ni^2+^ and Li^+^ was effectively prohibited during the cathode material synthesis, and no secondary phases or impurities peaks appear in the diffraction patterns, which agrees well with the literature [19]. 

Figure 2b shows the diffraction pattern of the graphite, which was used as an anode for the high energy/power lithium-ion batteries. The strongest diffraction peak at 26.46° corresponds to graphite (002) reflection. The crystal plane can be calculated according to the Bragg formula, in which the d002 spacing is 0.3365 nm and the characteristic diffraction peaks are at 43.35°, 44.46°, and 54.53°, which appear attributable to the crystal planes of 100, 101, and 004 in the graphite anode [20]. The graphitization degree *G* of the sample was calculated using the Mering–Maire equation (*G* = (0.3440 − d002)/(0.3440 − 0.3354) × 100%) to be about 87% [21].

Figure 3a–c shows the scanning electron microscope images of NCM523 material. The primary particles of NCM523 are elongated crystals ranging from 200 to 500 nm in width and 200 to 1000 nm in length. The particles are smooth, and the agglomeration is relatively tight, where the diameters of secondary particles are 3 to 4 µm.

Figure 3d–f displays the micrograph of the negative material graphite. The particles are mostly long strips 15 µm long and about 10 µm wide, in addition to some spherical particles 5 µm in diameter. The surface is smooth, showing traces of coating treatment, and layered stacking is vaguely visible. However, no obvious cross-section is exposed. In addition, Figure 4 shows the cross-sectional SEM, and Table 1 lists the component analysis. The atomic numerical results indicate that the materials are well synthesized.

The electrical characteristic curve of positive material NCM523 coin cell is shown in Figure 5a. The initial charge capacity is 200.2 mAh·g^−1^, and the discharge capacity is 178.4 mAh·g^−1^; thus, the initial coulombic efficiency is 89%. Figure 5b shows the charge–discharge curve of negative material graphite coin cell. The initial discharge capacity is 361.8 mAh·g^−1^, and the charge capacity is 340.5 mAh·g^−1^; thus, the initial coulombic efficiency is 94.1%.

Adjust the state of charge of the graphite anode from SOC1 to SOC7 and set the temperature of the oven to T1 to T3, then conduct a cross EIS test. The test results are shown in Figure 6. The spectra of negative electrodes at each temperature when SOC is adjusted to zero SOC exhibit similar characteristics. Over the test frequency range, the spectra of graphite can be divided into three parts—a ~45° line in the high-frequency region, a semicircle in the intermediate-frequency region, and an oblique line in the low-frequency region. When the electrode is fully charged to 100%, the spectral features do not change significantly.

According to Zhuang [22] and Li [23], the 45° line in the high-frequency region (HFR) relative to the real axis is characterized by the behavior of ion migration in the electrode with rich pore size. In addition, the incomplete semicircle in the middle frequency region (MFR) represents the electrochemical reaction resistance, which is a superposition of several semicircles, as proposed by the equivalent circuit below. Finally, the oblique lines in the low-frequency region (LFR) reflect the Li^+^ moving in the internals of the graphite solid material. The semicircle representing the SEI film resistance is not observed in the ultra-high frequency (UHF) range, which may be due to the low resistance value [24,25,26,27,28].

Adjust the state of charge of the NCM523 cathode from SOC1 to SOC7 and set the temperature of the oven to T1 to T3, then conduct a cross EIS test. The test results are shown in Figure 7. The EIS spectral characteristics of the NCM523 electrode can be divided into two parts at −20 °C and 0% SOC—a diagonal in the HFR and an arc in the LFR. The tangent at the curved line is nearly horizontal. As the temperature increases, the radius of curvature of the arc in the LFR decreases continuously.

When the over potential of the cathode is adjusted to SOC2 at −20 °C, the EIS spectral characteristics consist of a diagonal in the HFR and in the LFR the semicircle tending to be complete. Notably, the semicircle transforms into a semicircle in the mid-frequency region and a slanted line in the low-frequency region as the temperature increases. At this time, the EIS spectra feature consists of three parts—a ~45° line in the HFR relative to lateral axis, a curved line in the MFR, and a half circle at the LFR.

When the over potential of the cathode is adjusted from SOC1 to SOC7, spectral characteristics do not change significantly. A ~45° line from lateral axis in the HFR, the semicircle in the MFR, and the oblique line in the LFR can be attributed to the behavior of ion migration in the electrode with rich pore size, the electrochemical reaction resistance, and the Li^+^ moving in the internal of NCM523 solid material, respectively.

Combining the data characteristics of this study with the electrochemical reaction process, we proposed RC circuit model of cathode and anode in a symmetrical battery in Figure 8, where *R*_ohm_ is the resistance, which includes the barrier of electronics moving in the metal collector and the organic carbonate and *Z_W_* is the the barrier of lithium ions mitigating in the solid NCM523 or graphite (not discussed in this paper). The behavior of Li ion migration in the electrode with rich pore size with plenty of liquid electrolyte is symbolized by a uniform transmission line model element de Levie-Finite Length Pore (*L*_s_).

According to the theory of uniform transmission line model [15], the component *L_S_* is defined below:(1)$$Z=\sqrt{\frac{R_{ion}}{{\left(j\omega \right)}^{\alpha }Q_{dl}}}\mathrm{Cot}h\left(\sqrt{R_{\mathrm{ion}}\left({\left(j\omega \right)}^{\alpha }Q_{dl}+\frac{1}{R_{ct,A}}\right)}\right)$$
where *Z* is the complex impedance of the electrode (Ω); *j* is the imaginary unit; *ω* = 2 π *f*, where *f* is the frequency (Hz); (*jω*)^α^*Q_dl_*, where 0 < *α* ≤ 1 is a constant phase element [16,17,18], which is a substitute for the ideal double layer capacitor element *C_dl_*; *R*_ion_is the ionic resistance of the electrolyte in porous electrodes (Ω); and *R*_ct_ is the charge transfer resistance of the entire electrodes (Ω). Two limiting cases are generally taken into consideration. One case is for infinitely high frequencies where *Z* = 0 (from Equation (1), as *ω* → ∞, *Z* → 0), and another case is for the low frequencies (from Equation (1), as *ω* → 0). Therefore, *Z* is simplified as below:(2)$$L_{s}\equiv Z_{\omega \to 0}=\sqrt{R_{ion}\cdot R_{ct}}Coth\left(\sqrt{R_{ion}/R_{ct}}\right)$$

Regardless of how large the absolute value of the ionic resistance *R*_ion_ within the porous electrodes or the charge transfer resistance *R*_ct_, by approximation of the coth function, when *R*_ion_ ≪ *R*_ct_, *L_s_* is defined as the following:(3)$$L_{s}|{}_{\left(R_{ion}\langle \langle R_{\mathrm{ct}}\right)}=R_{\mathrm{ct}}+\frac{R_{\mathrm{ion}}}{3}$$

As the porous electrodes become thicker, another typical phenomenon is obvious. Especially at low temperatures, *R*_ion_ ≫ *R*_ct_, and the electrochemical reaction controlling factor drops in the transport limited regime, i.e., the ratio *R*_ct_/*R*_ion_ becomes very large. As a result, Equation (2) simplifies to the following expression:(4)$$L_{s}|{}_{\left(R_{ion}\rangle \rangle R_{\mathrm{ct}}\right)}=\sqrt{R_{\mathrm{ct}}\cdot R_{\mathrm{ion}}}$$

The characteristics of the impedance spectra correspond to Equation (1), where the ~45° line is between the high-frequency region and the real axis and a semicircle is in the low-frequency region. Therefore, the equivalent circuit corresponding to Equation (1) is shown in Figure 8, which better fits the EIS experimental data, and its data comparison chart is shown in Figure 9. Additionally, the experimental data curve and the fitted curve achieve a good overlap.

Figure 10a shows the *R*_ion_ change of anode with the SOC under temperature series. Clearly, *R*_ion_ decreases rapidly between −20 and 25 °C, from a maximum of 90 Ω at −20 °C to less than 20 Ω at 25 °C. This phenomenon is caused by many factors, as studied by many researchers. The main cause is attributed to the electrolyte conductivity quickly decreasing as temperature decreases. In addition, *R*_ion_ first increases and then decreases, going from 10–100% SOC. Figure 10b shows the *R*_ct_ change of the graphite electrode with the SOC at different temperatures, in which a weak law of first decreasing and then increasing is evident, minimizing at 50% SOC. The expression of the *R*_ct_ is as follows [23]:(5)$$R_{ct}=\frac{RT}{n^{2}F^{2}c_{\max}k_{0}{\left(M_{Li^{+}}\right)}^{\left(1-\alpha \right)}{\left(1-x\right)}^{1-\alpha }x^{\alpha }}$$
where *R*, *T*, *F* are all constants. *n* is reaction electrons number; *M*_Li_^+^ is ion concentration in the electrolyte; *x* is the degree of lithium insertion which is usually another similar representation method for SOC; *k*_0_ is fixed once the reaction process is clear; although *α* as one factor but usually 0.5. Equation (5) indicates that *R*_ct_ first decreases, then increases when *x* strat from 0, *R*_ct_ will reach the valley of the continuous changing when *x* is ~0.5. Obviously, as the state of charge changes, corresponding *R*_ct_ change trend conforms to the result indicated by Equation (5), further confirming that the equivalent circuit selected is reasonable and the attribution of each impedance spectra feature is appropriate. Furthermore, the changing tendency of *R*_ct_ deriving from graphite follows Equation (5). Notably, *M*_Li_^+^ decreases dramatically with the temperature drop. The increase in *M*_Li_^+^ is much greater than the decrease in *T* in the numerator; thus, *R*_ct_ increases.

Figure 11 shows the variation of *R*_ion_ and *R*_ct_ with the SOC of the NCM523 electrode at different temperatures. The variation of *R*_ion_ of the NCM523 electrode with the SOC at different temperatures is similar to that of the graphite electrode, but its value is much lower. In general, at each temperature, the *R*_ct_ of the NCM523 electrode first decreases and then increases with the SOC, also aligning with Equation (5). In other words, to realize fast charging and discharging, the working SOC of the NCM523 electrode should be controlled at around 50% SOC, especially at low temperatures.

Of note, *R*_ct_ and *R*_ion_ usually exist simultaneously, which are the key kinetic parameters for porous electrodes. To differentiate which contributes more to the electrochemical reactions under different conditions, a factor is defined, § ≡ *R*_ct_/*R*_ion_, and Equation (2) can be expressed as follows:(6)$$\frac{L_{s}}{R_{ion}}=\sqrt{\xi }\mathrm{Cot}h\left(1/\sqrt{\xi }\right)$$

In order to quantitatively evaluate the speed control steps affecting the NCM523 HEP LIB resistance in the SOC range and temperature range, cluster analysis is conducted for *R*_ion_ and *R*_ct_. Figure 12a shows the relationship between § and 1000 *T*^−1^ for graphite electrodes, where § = 1 indicates that the absolute values of *R*_ion_ and *R*_ct_ are equivalent. When the temperature satisfies 1000 *T*^−1^ > 3.66 (i.e., *T* < 0 °C), the *R*_ct_ of the graphite electrode is much higher than *R*_ion_ in any SOC, indicating that the source of resistance (*R*_ion_ + *R*_ct_) is mainly *R*_ct_. When the temperature satisfies 3.66 > 1000 *T*^−1^ > 3.53 (i.e., *T* = 10 to 0 °C), the curves of 1/*R*_ion_ with the temperature term 1000 *T*^−1^ and 1/*R*_ct_ with 1000 *T*^−1^ overlap, and the two resistance values are nearly equal, indicating that *R*_ion_ and *R*_ct_ have the same resistance at this temperature.

The improvement of liquid diffusion in the electrode pores and charge transfer for battery resistance design are all crucial. When § > 1, adjusting *R*_ct_ to improve quick electric performance of anode is a clear solution. Considering the results indicated by Equations (2) and (3), *R*_ion_ is proportional to the electrode thickness, contrary *R*_ct_ is much larger when the electrode much thicker. Thus, once the electrode manufacture processing is fixed, *R*_ion_ and *R*_ct_ are usually just minor changes, optimizing the coating weight of the electrode to obtain a much smaller internal resistance of the anode is critical, especially in extreme environments. Moreover, improving the micro-structure of electrode, electrode preparation process conditions, and electrolyte composition and reducing *R*_ion_ and *R*_ct_ are obviously the fundamental means to improve the electrode charge–discharge rate performance.

Figure 12b shows the relationship between § and 1000 *T*^−1^ of the NCM523 electrode. Clearly, the § curves under different SOCs are intertwined, which is similar to the graphite electrode behavior, but § is greater than 1. Values of § >1 indicate that the effect of *R*_ion_ on the NCM523 cathode is not significant, and *R*_ct_ is the main factor affecting the reaction kinetics in the whole process. When the temperature satisfies 1000 *T*^−1^ > 3.66 (i.e., *T* = 0 °C), § (where §_min_ > 20 except at 50% SOC) is significantly higher than the absolute value of the graphite electrode § (where §_max_ < 20). Because *R*_ct_ is usually fixed, once the active material is selected, § is much smaller, and *R*_ion_ is much larger. In addition, *R*_ion_ is easily changed by the optimization of the industrialization process. A smaller graphite § indicates that improving the resistance design of the graphite electrode at low temperature is more effective in improving battery power performance. On the other hand, to improve the dynamics of NCM523 electrode, it is necessary to focus on improving the intrinsic characteristics of the electrochemical reaction resistance of the material, such as electronic conductivity and electronic conductivity.

Benefiting from the collection of data from multiple temperature points, we can further analyze the effect of temperature on *R*_ion_ and *R*_ct_, the Arrhenius formula is proposed, as shown in Equation (7). Activation energies for *R*_ion_ and *R*_ct_ are defined as the following:(7)$$\frac{1}{R_{x}}=A\exp\left(\frac{-E_{a}}{RT}\right)$$

Linear fitting data are shown in Figure 13. Here, *A* is the pre-index factor; *R* is a constant of the properties of ideal gases; *T* is absolute temperature; *E_a_* indicates the minimum energy required by the reactant molecule to reach the activated molecule. *E*_a_ of *R*_ion_ and *R*_ct_ for the NCM523 and graphite electrodes keep stable with the state of charge, and they are also very close in size, respectively. The activation energy of *R*_ion_ is 15 to 30 kJ·mol^−1^, and the activation energy of *R*_ct_ is 60 to 80 kJ·mol^−1^, these values are basically close to the test results of other methods [29]. The activation energy of *R*_ct_ is two to four times higher than that of *R*_ion_, indicating that the energy required to trigger *R*_ct_ change is much higher than *R*_ion_; at the same time, the benefits of battery performance improvement after improving R are also greater.

## 4. Conclusions

In summary, the impedance spectra characteristics of the graphite and NCM523 electrodes of the HEP LIB under series of temperatures and SOC range were studied using EIS technology in a symmetric battery system. The impedance spectra data were fitted, and the lithium-ion liquid-phase diffusion resistance *R*_ion_ and charge transfer resistance *R*_ct_ inside the porous electrode were successfully distinguished. In addition, this work introduces the quantitative parameter § ≡ *R*_ct_/*R*_ion_, and § = 1 is the benchmark to discuss the lithium-ion liquid-phase diffusion and charge transfer processes inside the porous electrode as the boundary conditions of the rate control step. The performance improvement of the ion power battery, especially the high energy/power lithium-ion batteries, in specific temperatures and SOC ranges directed the design. Moreover, the *R*_ion_ and *R*_ct_ activation energies of the cathode and anode electrodes were measured, the fitting result of activation energy representing the difficulty of *R*_ct_ was 60 to 80 kJ·mol^−1^, and the activation energy of *R*_ion_ was 15 to 30 kJ·mol^−1^. Therefore, the analysis results of the variation of *R*_ion_ and *R*_ct_ with temperature and the SOC show that improving the charge–discharge rate of graphite electrodes is more effective than improving the battery power performance at low temperatures.

## Figures and Tables

**Figure 1 nanomaterials-13-00856-f001:**
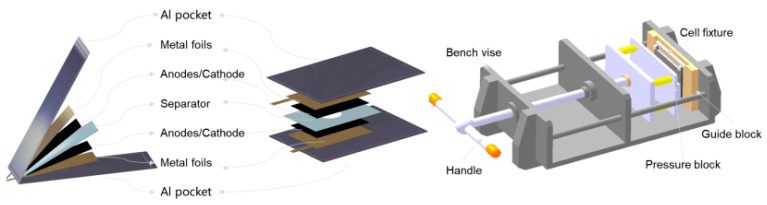
Schematic diagram of assembly and testing of the symmetric cells.

**Figure 2 nanomaterials-13-00856-f002:**
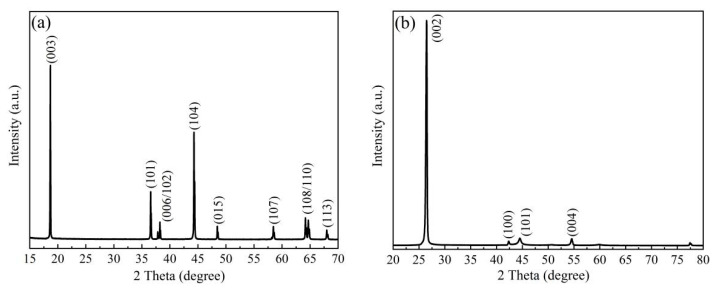
X-ray diffraction spectrum of (**a**) NCM523 and (**b**) graphite.

**Figure 3 nanomaterials-13-00856-f003:**
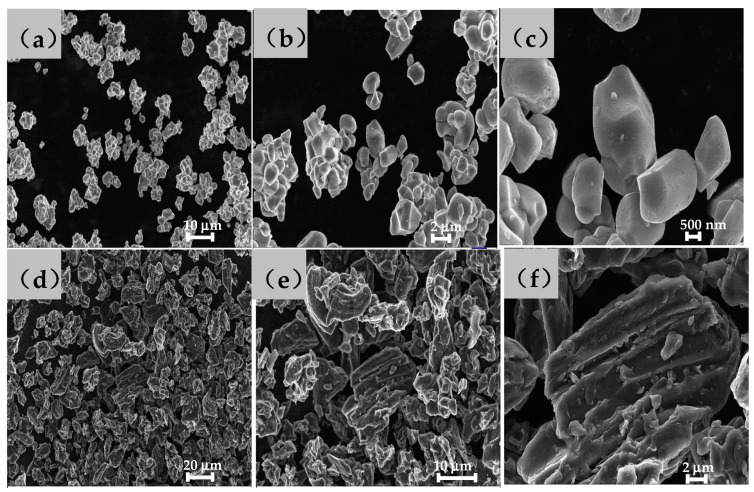
Morphology of (**a**–**c**) NCM523 and (**d**–**f**) graphite materials.

**Figure 4 nanomaterials-13-00856-f004:**
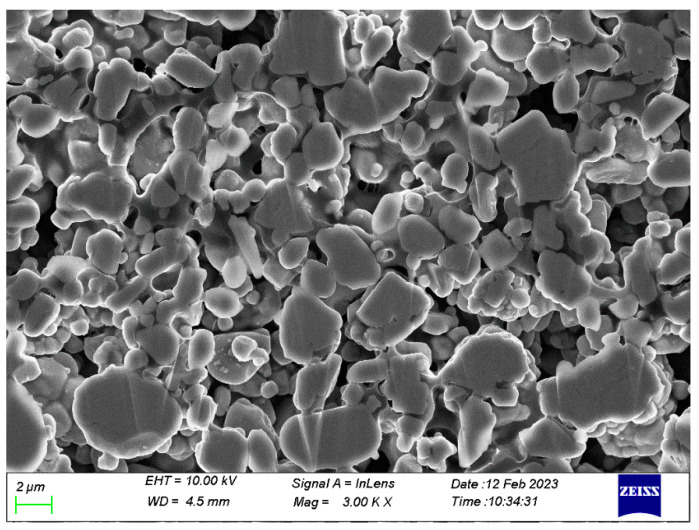
Cross-sectional morphology of NCM523.

**Figure 5 nanomaterials-13-00856-f005:**
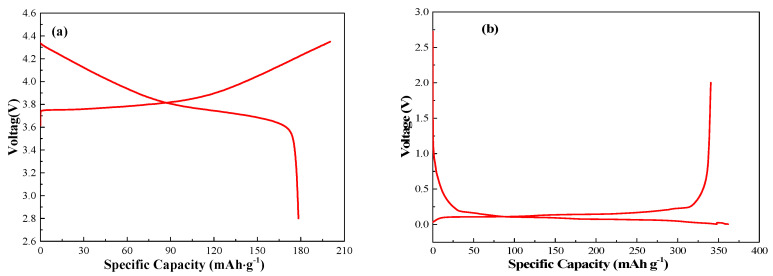
Charge/discharge curves of the (**a**) NCM523 and (**b**) graphite electrodes.

**Figure 6 nanomaterials-13-00856-f006:**
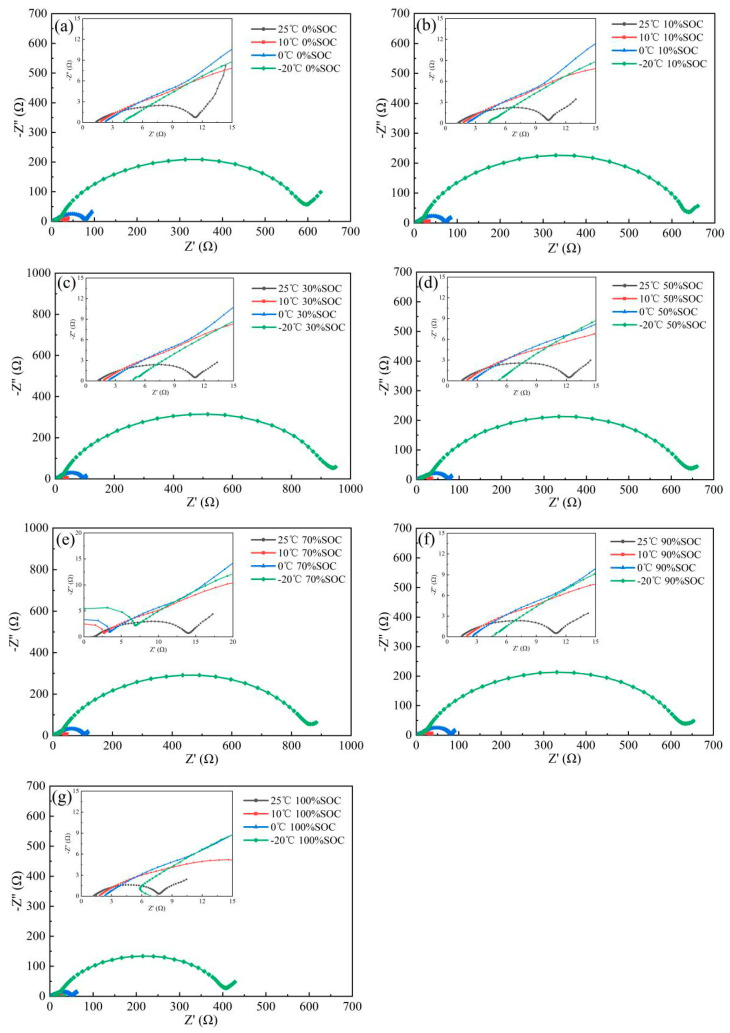
EIS spectrum of the graphite (**a**) 0% SOC; (**b**)10% SOC; (**c**) 30% SOC; (**d**) 50% SOC; (**e**) 70% SOC; (**f**) 90% SOC; (**g**) 100% SOC.

**Figure 7 nanomaterials-13-00856-f007:**
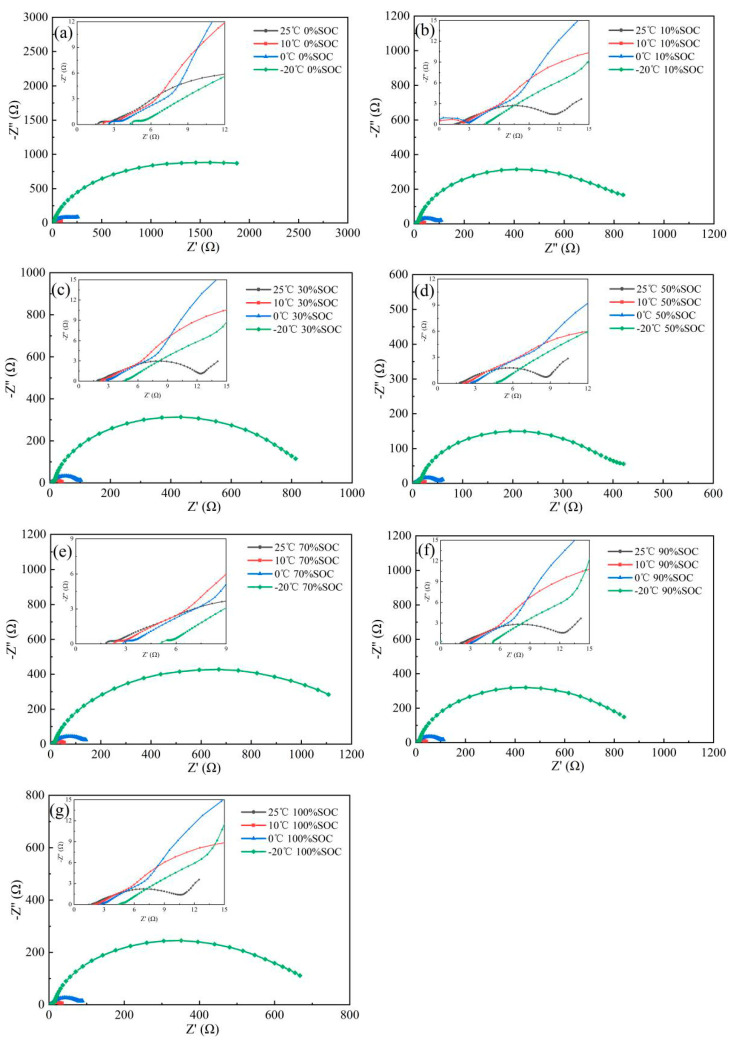
EIS spectrum of the NCM523 (**a**) 0% SOC; (**b**)10% SOC; (**c**) 30% SOC; (**d**) 50% SOC; (**e**) 70% SOC; (**f**) 90% SOC; (**g**) 100% SOC.

**Figure 8 nanomaterials-13-00856-f008:**
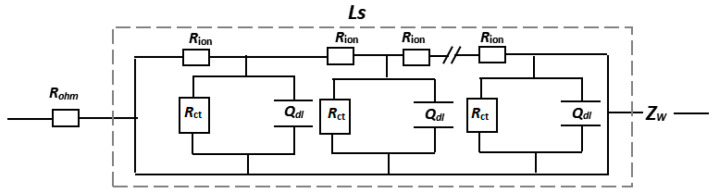
RC circuit suggested for symmetric cells for NCM523 or graphite.

**Figure 9 nanomaterials-13-00856-f009:**
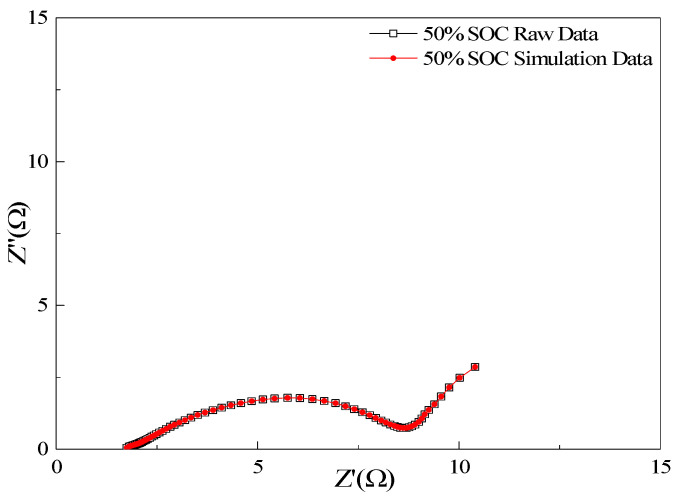
Comparison between testing data and fitting result.

**Figure 10 nanomaterials-13-00856-f010:**
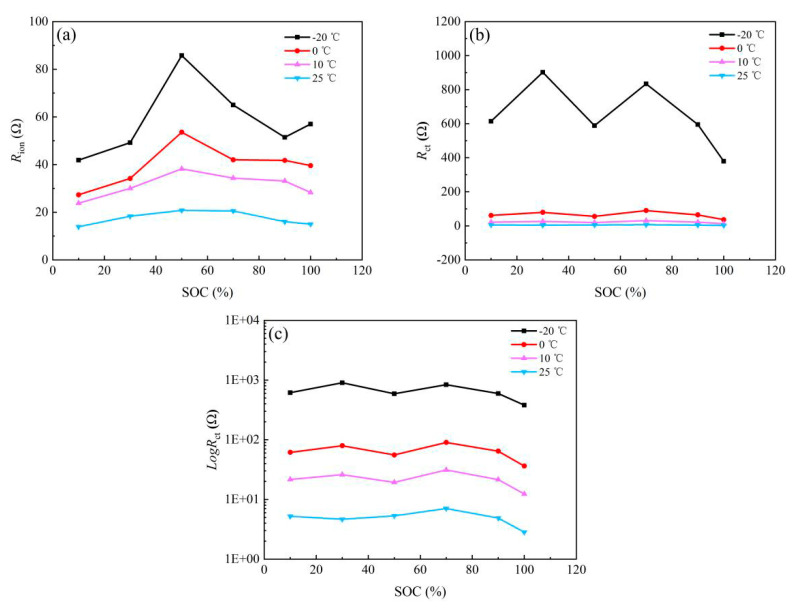
Plotting of anode in (**a**) *R*_ion_ and (**b**) *R*_ct_ (**c**) Log*R*_ct_.

**Figure 11 nanomaterials-13-00856-f011:**
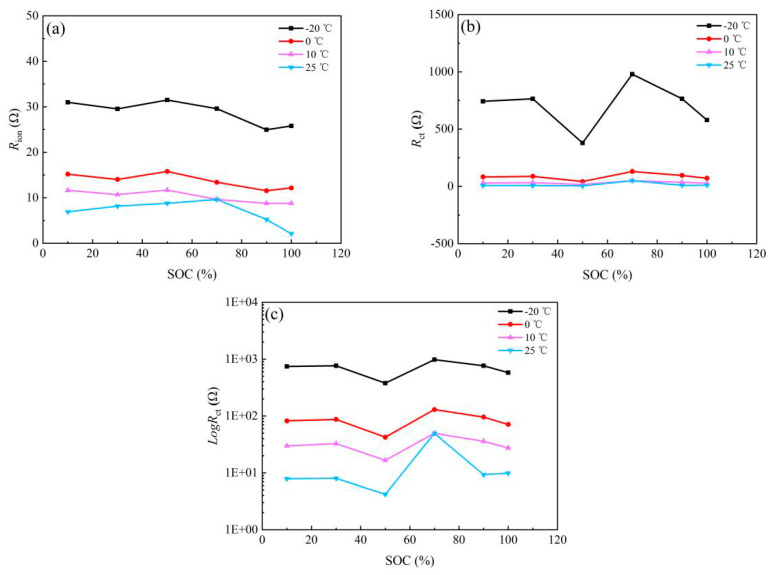
Plotting of cathode in (**a**) *R*_ion_ and (**b**) *R*_ct_ (**c**) Log*R*_ct_.

**Figure 12 nanomaterials-13-00856-f012:**
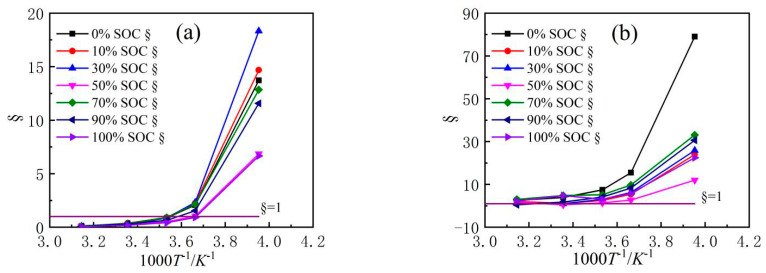
Variations of the § with 1000.T^−1^ (**a**) Anode (**b**) Cathode.

**Figure 13 nanomaterials-13-00856-f013:**
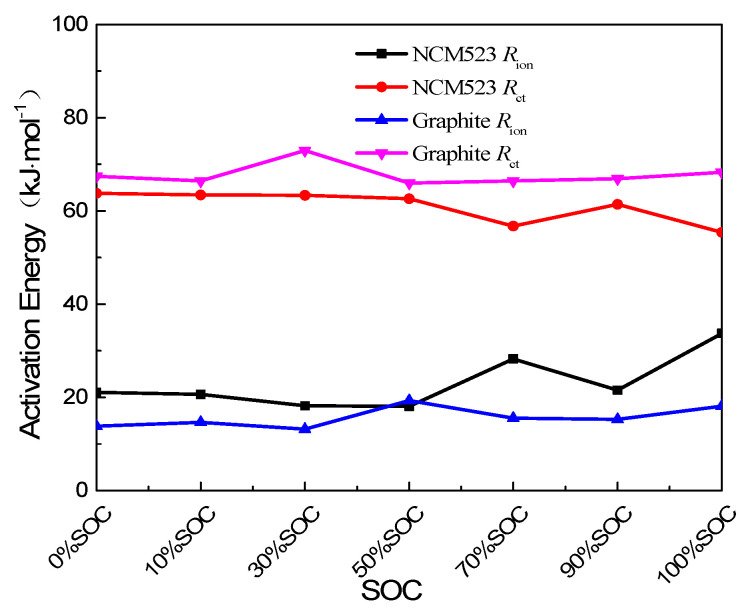
Activation energy of *R*_ion_ and *R*_ct_ both for cathode and anode.

**Table 1 nanomaterials-13-00856-t001:** The compositional raio of NCM523 cathode material.

Eelement	Line Type	wt%	wt% Sigma	Atomic Percentage	Standard Sample Label	Manufacturer Standard
C	K Line	13.27	0.31	26.53	C Vit	Yes
O		34.41	0.22	51.66	SiO_2_	Yes
Mn		15.27	0.14	6.68	Mn	Yes
Co		10.66	0.16	4.34	Co	Yes
Ni		26.39	0.23	10.79	Ni	No
Sum:		100.00		100.00		

## Data Availability

Data will be made available on request.

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
