# Peer review of "Ion Transport and Electrochemical Reaction in LiNi0.5Co0.2Mn0.3O2-Based High Energy/Power Lithium-Ion Batteries"

_nanomaterials, 2023, doi:10.3390/nano13050856_

Round 1

Reviewer 1 Report

The authors studied the interfacial reaction kinetics by EIS. There are many equations and experimental results to evaluate the kinetics, but several aspects should be addressed before taking further steps.

1.       Overall, there are many typos in the paper. It's better to check the manuscript.

2.       Figures 5 and 6 show the tendency of Rct to decrease depending on SOC. However, why was a smaller impedance observed at 50% SOC than 70% SOC.

3.       In the manuscript, the description of the reason why impedance increase according to temperature under the equal SOC conditions is insufficient. In other words, the author should explain why Rb and Rct increase with lower temperature.

Author Response

Reviewer #1 (Remarks to the Author):

The authors studied the interfacial reaction kinetics by EIS. There are manyequations and experimental results to evaluate the kinetics, but several aspects should be addressed before taking further steps.

Reply: Thanks the reviewer for the valuable suggestions.

  1. Overall, there are many typos in the paper. It's better to check the manuscript.

Reply: Thank the reviewer for carefully and patiently reviewing and correcting.  We have revised the manuscript in a professional language polishing agency. Please see more details highlighted in yellow background in the revised manuscript.

  1. Figures 5 and 6 show the tendency of Rct to decrease depending on SOC. However, why was a smaller impedance observed at 50% SOC than 70% SOC.

Reply: Thank the reviewer for pointing out the minor change. The expression of the charge transfer resistance Rct is as follows [23]:

                                   (5)

In the formula, the parameter of R, T, n, F ,Cmax, K and MLi+ are constant, Cmax and α are basically fixed especially α=0.5 usually. Therefore, the minimum value Rct  is always at the point of 50%SOC. About this point, We have already cited in the revised manuscript and marked with yellow background

Reference:

[23] Li, Y.; Wu, H. Theoretical treatment of kinetics of intercalation electrode reaction. Electrochim. Acta 1989, 34, 157-159.

  1. In the manuscript, the description of the reason why impedance increase according to temperature under the equal SOC conditions is insufficient. In other words, the author should explain why Rb and Rct increase with lower temperature.

Reply: We thank the reviewer for the good suggestions. We have added a description that accompanies with Fig9 and Fig10. in the section 3, which have marked with the yellow background in the revised manuscript. The specific description as below:

This phenomenon is caused by many factors and has been studied by many researchers. The main cause is attributed to the electrolyte conductivity quickly decreasing with temperature decrease.

[2] Zhang, N.; Deng, T.; Zhang, S. Q.; Wang, C. H.; Chen, L. X.; Chunsheng Wang, C. S.; Fan, X. L. Critical review on low-temperature Li-ion/metal batteries. doi: 10.1002/adma.202107899.

Rct changing trend follows the Formula 5, MLi+ is the concentration of lithium ions in the solution, and it is in the denominator of the formula. MLi+ decreases dramatically with the temperature drop. The value of the increase of MLi+ is much greater than the value of the decrease of numerator position T, resulting in the increases of Rct.

Reviewer 2 Report

This paper reports the study The energy-power hybrid lithium-ion batteries using LiNi0.5Co0.2Mn0.3O2 has an excellent trade-off between specific capacity, cost, and thermal stable characteristics, whereas it is still 13 a huge challenge for power improvement under low temperatures. Deeply understanding of the electrode interface reaction mechanism is an important basis for solving this problem. In this work, the impedance spectrum characteristics of commercial symmetric batteries are studied under different SOC and temperatures. The changing tendencies of Li+ diffusion resistance Rion and charge transfer resistance Rct with temperature and SOC are explored. Moreover, one quantitative parameter § ≡ Rct/Rion is introduced to identify the boundary conditions of the rate control step inside the porous electrode. This work points out the direction to design and improve performance for commercial energy-power hybrid lithium-ion power batteries with specific temperatures and SOC intervals.  The review results are interesting and informative, but some data were not well presented. Details are listed below.

The author synthesized a catalyst of La-doped Ag3PO4 and conducted photocatalytic experiments. The aurthors calculated the size of La-doped AgNO3 using Sherrer’s equation in XRD data. However, it is not possible to calculate the exact mean size and standard distribution using the formula calculated by Sherrer’s equation. Therefore, it is necessary to the exact size though SEM and indicate the standard distribution.

1)      The authors synthesized the LiNi0.5Co0.2Mn0.3O2 catalyst and analyzed it with SEM and TEM. However, to confirm the correct alloy structure, cross-sectional compositional line profiles must be added.

2)      In order to confirm the stability of the catalyst after electrochemical stability, SEM and TEM images after electrochemical stability experiments should be added.

Author Response

Reviewer #2 (Remarks to the Author):

This paper reports the study The High Energy/power Lithium-ion Batteries using LiNi0.5Co0.2Mn0.3O2 has an excellent trade-off between specific capacity, cost, and thermal stable characteristics, whereas it is still 13 a huge challenge for power improvement under low temperatures. Deeply understanding of the electrode interface reaction mechanism is an important basis for solving this problem. In this work, the impedance spectrum characteristics of commercial symmetric batteries are studied under different SOC and temperatures. The changing tendencies of Li+ diffusion resistance Rion and charge transfer resistance Rct with temperature and SOC are explored. Moreover, one quantitative parameter § ≡ Rct/Rion is introduced to identify the boundary conditions of the rate control step inside the porous electrode. This work points out the direction to design and improve performance for commercial high energy/power lithium-ion batteries with specific temperatures and SOC intervals. The review results are interesting and informative, but some data were not well presented. Details are listed below.

Reply: Thanks to the reviewer for his professional and helpful guidance, which is valuable for us to improve the manuscript significantly. We wish that the reviewer finds the updated version of the manuscript to be suitable for its publication in Nanomaterials.

  1. The authors synthesized the LiNi5Co0.2Mn0.3O2 catalyst and analyzed it with SEM and TEM. However, to confirm the correct alloy structure, cross-sectional compositional line profiles must be added.

Reply: Thank you for your valuable suggestions. The material LiNi0.5Co0.2Mn0.3O2 catalyst used in this study is provided by Tianjin Bamo Technology Co., Ltd. (Added in the manuscript with yellow background), it’s commercial cathode material and is already in circulation on the market. As the reviewer required, we have attached the cross-sectional SEM and component analysis in table 1. The atomic percentage shows that the material is well synthesized.

Table 1. The compositional ratio of NCM523 cathode material.

Eelement

Line type

wt%

wt% Sigma

atomic percentage

Standard sample label

Manufacturer standard

C

K Line

13.27

0.31

26.53

C Vit

Yes

O

34.41

0.22

51.66

SiO2

Yes

Mn

15.27

0.14

6.68

Mn

Yes

Co

10.66

0.16

4.34

Co

Yes

Ni

26.39

0.23

10.79

Ni

No

Sum:

100.00

100.00

  1. In order to confirm the stability of the catalyst after electrochemical stability, SEM and TEM images after electrochemical stability experiments should be added.

Reply: Thank the reviewer for suggestions about the consideration of long-term stability. It is worth noting that both the research electrode and the counter electrode were energy-type lithium ion power battery NCM523 or graphite electrodes under the fresh status which means they just perform one charge and discharge cycle and then be sent to do disassembly. Therefore, the durability test was not performed in our work. Of course, in order to make readers understand more clearly, the relevant positions in this article are also supplemented with yellow background.

Reviewer 3 Report

This paper is devoted to a study on the interfacial reaction kinetics by electrochemical impedance spectroscopy in commercial samples of power hybrid lithium-ion batteries. The latter represents the special value of the paper making it worthy for publication. However, there are still a number of insignificant shortages requiring revision.
1. The current density has a dimension of A/m2. The authors give a charge amount as measure of current density. A charge amount is in turn a measure of current. For an estimation of current density the area through which the charges flow must be known. The paper does not explain the conversion of given charge amount into a current density. In this context, I cannot understand how the relation 1 C = 1,602 176 634 · 10-19 As corresponds to 37 A.
2. The authors should explain more in detail that the I003/I104 peak ratio is a  measure of the degree of cation mixing of Li+/Ni2+, in which a value higher  than 1.2 indicates that a well-formed layered structure dominantes over a cubic rock-salt Ni2+O structure as a result of the cation ordering of Li+/Ni2+. Moreover, they should provide a reference for interested reader convinience.
3. A reference to the Mering-Maire equation is required.
4. The interpretation of the impedance spectrum needs revision. The semicircle in the intermediate frequency region is more a superposition of several semicircles as proposed by the equivalent circuit in Figure 7. This is a hint to a Gerisher element obtained, e.g., for porous La0.7Ca0.3Cr0.2Ti 0.8O3-d SOFC anodes [DOI: 10.1002/1615-6854(200112)1:3/4<256::AID-FUCE256>3.0.CO;2-I].
5. The melting point of Li is 180°C. Where Li diffusion in the liquid phase arise from ?
6. In order to fit the rules of notation of physical values, the ratio Rct/Rion is better denoted by a creek lowercase letter.
7. In pages 2, 3, 4, 7 and 8 "~" is better replaced by "..." or "to"
8. The derived activation energies of Rct and Rion should be compared with literature data of similar compounds.

Author Response

Reviewer #3 (Remarks to the Author):

This paper is devoted to a study on the interfacial reaction kinetics by electrochemical impedance spectroscopy in commercial samples of high energy/power lithium-ion batteries. The latter represents the special value of the paper making it worthy for publication. However, there are still a number of insignificant shortages requiring revision.

Reply: Thanks to the reviewer for your affirmation and great attention to details. I hope that the corresponding instructions and supplements can completely and fully understand your intentions, and the revised manuscript is suitable for publication in Nanomaterials.

  1. The current density has a dimension of A/m2. The authors give a charge amount as measure of current density. A charge amount is in turn a measure of current. For an estimation of current density the area through which the charges flow must be known. The paper does not explain the conversion of given charge amount into a current density. In this context, I cannot understand how the relation 1 C = 1,602 176 634 10-19 As corresponds to 37 A.

Reply: Thanks to the reviewer for the question. In the full battery, the current densities of SOC were regulated by SOC at 0.1C (1C is equal to 37 A, 37A is calculated from the total effective area of cathode electrode 2.6 m2 * coating weight 8.3mg·cm-2 * active material ratio 96 wt.% * specific capacity test in coin cell 178.4 mAh·g-1 ). To better understand, the descriptions have already added in the section 2 in revised manuscript.

   Theoretical specific capacity of material calculation formula: mAh·g-1 = 1/M*Li measurement number*F/t. In this formula, M is Molar mass of material. F is Faraday's constant F = eNA = 96485 C·mol-1, F=9.65×10000 C·mol-1, N is Avogadro number N=6.02214×1023 mol-1, e is elementary charge e=1.602176·10-19 C, t is time 3600s. In this article, it is not recommended to list the basic formula anymore.

  1. The authors should explain more in detail that the I003/I104 peak ratio is a measure of the degree of cation mixing of Li+/Ni2+, in which a value higher than 1.2 indicates that a well-formed layered structure dominantes over a cubic rock-salt Ni2+ O structure as a result of the cation ordering of Li+/Ni2+. Moreover, they should provide a reference for interested reader convince.

Reply: Thank you for your valuable suggestions. Regarding the ratio of the intensity of the (003) diffraction peak to the (104) diffraction peak of the layered cathode material, this ratio index is actually a semi-quantitative and semi-empirical measure of the cationic disorder. The conclusion is supported by below data analysis. We have cited literature and revised the manuscript with yellow background.

[3] Wang, F.; Zhang, Y.; Zou, J.Z.; Liu, W.J.; Chen, Y.P.The structural mechanism of the improved electrochemical performances resulted from sintering atmosphere for LiNi0.5Co0.2Mn0.3O2 cathode material. Journal of Alloys and Compounds. 2013, 558,172-178.

  1. A reference to the Mering-Maire equation is required.

Reply: Thanks to the reviewer for the question. The reference of Ref. [21] about the Mering-Maire equation has already cited and marked with yellow background.

G=(0.3440–d002)/(0.3440–0.3354)×100%, G is degree of graphitization.

[21] Qiu T, Yang J-g, Bai X-j. Insight into the change in carbon structure and thermodynamics during anthracite transformation into graphite[J]. International Journal of Minerals Metallurgy and Materials, 2020, 27(2): 162-172.

  1. The interpretation of the impedance spectrum needs revision. The semicircle in the intermediate frequency region is more a superposition of several semicircles as proposed by the equivalent circuit in Figure 7. This is a hint to a Gerisher element obtained, e.g., for porous La7Ca0.3Cr0.2Ti0.8O3-d SOFC anodes [DOI: 10.1002/1615-6854(200112)1:3/4<256::AID-FUCE256>3.0.CO;2-I].

Reply: Thank the reviewer for instruction and correction. The interpretation of the impedance spectrum has been revised as below, and highlighted with yellow background in the manuscript.

Modification1: the semicircle in the intermediate frequency region is related to the charge transfer process which is a superposition of several semicircles as proposed by the equivalent circuit in Figure 8.

Modification2: Therefore, the oblique line approximately 45° from the real axis in the high frequency region, the semicircle in the intermediate frequency region, and the oblique line in the low frequency region can be attributed separately to the liquid-phase diffusion inside the porous electrode, several semicircles superposition of charge transfer, and solid-state diffusion of lithium ions.

  1. The melting point of Li is 180° Where Li diffusion in the liquid phase arise from ?

Reply: We appreciate the reviewer for the careful review of the manuscript. As you said, the accurate word is Li+ or lithium-ion, not Li, it is easy to mislead. In this study, we focus on the lithium-ion diffusion in an organic solvent which absorbed in the multiple pores of electrodes. In order to accurately express, we have replaced all the word Li with Li+ or lithium-ion, which has marked with yellow background in revised manuscript.

  1. In order to fit the rules of notation of physical values, the ratio Rct/Rion is better denoted by a creek lowercase letter.

Reply: We accept the reviewer’s opinion, which has already revised to creek lowercase symbol ξ with yellow background. Thanks for your good suggestion and it really looks much more academic.

  1. In pages 2, 3, 4, 7 and 8 "~" is better replaced by "..." or "to"

Reply: Thank you for your valuable suggestions. It has already revised with yellow background, which looks much more comfortable in revised manuscript.

  1. The derived activation energies of Rct and Rion should be compared with literature data of similar compounds.

Reply: Thanks to the reviewer for the question. The reference of Ref. [29] about the activation energy comparison has already cited and marked with yellow background.

The activation energy of Rion is 15 to 30 KJ·mol-1, and the activation energy of Rct is 60 to 80 KJ·mol-1, which are equivalent to those measured by AC impedance in the literature [29].

[29] Wei, T.; Zhuang, Q.C.; Wu, C.; Cui, Y.L.; Fang, L.A.; Sun, S.G. Effects of temperature on the intercalation-deintercalation process of lithium ion in the spinel LiMn2O4. Acta Chim. Sinica 2010, 68, 1481-1486.

Reviewer 4 Report

The article titled "Study on the Interfacial Reaction Kinetics by Electrochemical Impedance Spectroscopy for LiNi0.5Co0.2Mn0.3O2-based Energy-power Hybrid Lithium-ion batteries" written by J. Xu is very interesting and presents the utilization of very common material LiNi0.5Co0.2Mn0.3O2 used in commercial LiBs. I would like to believe that the article will find a lot of readers. In this regard, Introduction section should be improved with references where the informations about the global market results are included as well as with the explanaition of the presented problem. At the end, the English should be improved. I recommend the article to be published in the Nanomaterials MDPI journal after minor revision.

Author Response

Reviewer #4 (Remarks to the Author):

The article titled "Study on the Interfacial Reaction Kinetics by Electrochemical Impedance Spectroscopy for LiNi0.5Co0.2Mn0.3O2-based Energy-power Hybrid Lithium-ion batteries" written by J. Xu is very interesting and presents the utilization of very common material LiNi0.5Co0.2Mn0.3O2 used in commercial LiBs. I would like to believe that the article will find a lot of readers. In this regard, Introduction section should be improved with references where the informations about the global market results are included as well as with the explanaition of the presented problem. At the end, the English should be improved. I recommend the article to be published in the Nanomaterials MDPI journal after minor revision.

Reply: Thank you for your valuable suggestions. We have revised the manuscript in a professional language polishing agency. We have checked the revised manuscript thoroughly and carefully to avoid potential spelling, grammar and other mistakes. Please see more details highlighted in yellow background in the revised manuscript.

Round 2

Reviewer 2 Report

good!